# DeblurDiNAT: A Compact Model with Exceptional Generalization and Visual Fidelity on Unseen Domains

## Abstract

Recent deblurring networks have effectively restored clear images from the blurred ones. However, they often struggle with generalization to unknown domains. Moreover, these models typically focus on distortion metrics such as PSNR and SSIM, neglecting the critical aspect of metrics aligned with human perception. To address these limitations, we propose DeblurDiNAT, a deblurring Transformer based on Dilated Neighborhood Attention. First, DeblurDiNAT employs an alternating dilation factor paradigm to capture both local and global blurred patterns, enhancing generalization and perceptual clarity. Second, a local cross-channel learner aids the Transformer block to understand the short-range relationships between adjacent channels. Additionally, we present a linear feed-forward network with a simple while effective design. Finally, a dual-stage feature fusion module is introduced as an alternative to the existing approach, which efficiently process multi-scale visual information across network levels. Compared to state-of-the-art models, our compact DeblurDiNAT demonstrates superior generalization capabilities and achieves remarkable performance in perceptual metrics, while maintaining a favorable model size.

## 1 Introduction

Image deblurring aims to restore clarity from blurred images (Cho et al., 2012; Yan et al., 2017). Although non-deep learning methods have shown effectiveness under certain conditions (Chen et al., 2011; Bahat et al., 2017; Hirsch et al., 2011; Xu & Jia, 2010), their performance often falls short in real-world scenarios where blurred patterns exhibit various orientations and magnitudes (Zhang et al., 2022).

With the advancement of deep learning, convolutional neural networks (CNNs) have been widely employed to address image deblurring tasks (Kupyn et al., 2018; Tao et al., 2018; Purohit & Rajagopalan, 2020; Cho et al., 2021; Zou et al., 2021; Ji et al., 2022; Fu et al., 2022). To achieve a large receptive field, latest deep CNNs (Chen et al., 2022; Fang et al., 2023) are scaled to large model sizes. More recently, Transformers (Vaswani, 2017) have dominated computer vision tasks (Dosovitskiy et al., 2020; Liu et al., 2021; Yuan et al., 2021; Jiang et al., 2021). A key feature of Transformers is their self-attention (SA) mechanism, which inherently enables long-range feature learning. To help Transformer architectures capture local blurred patterns, several methods incorporating window partitioning and 2-D convolutions have been proposed (Wang et al., 2022; Tsai et al., 2022; Zamir et al., 2022; Kong et al., 2023). However, these studies often overlook short-range cross-channel interactions, leading to channel-wise information loss. This limitation hampers the networks' generalization capabilities and perceptual performance, both of which are crucial for effectively addressing real-world image deblurring challenges.

We begin by discussing the **limited generalization problem** in state-of-the-art (SOTA) deblurring approaches. A typical manifestation of this issue is **mode collapse** in image deblurring tasks, as illustrated in Figure 1. While UFPNet (Fang et al., 2023) and FFTformer (Kong et al., 2023) achieve SOTA performance on benchmark datasets, their generalization to real-world blurred images remains questionable. Both models produce undesirable noisy pixels regardless of the input, resulting in their inability to adapt effectively to unseen domains. Notably, such an issue has not been observed in the deblurred results of our proposed DeblurDiNAT.

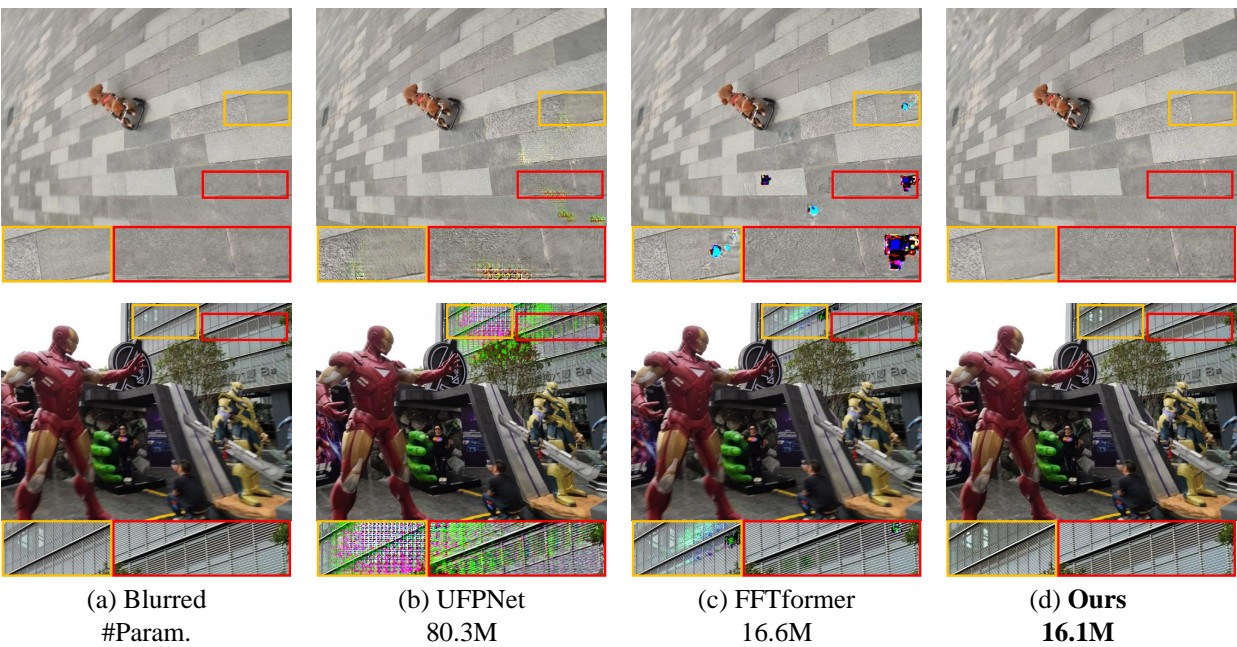

(a) Blurred
#Param.

(b) UFPNet
80.3M

(c) FFTformer
16.6M

(d) **Ours**
**16.1M**

Figure 1: Deblurred results on the unseen RWBI dataset (Zhang et al., 2020), where **mode collapse** occurs in SOTA networks, UFPNet (Fang et al., 2023) and FFTformer (Kong et al., 2023). In contrast, our proposed DeblurDiNAT resolves this issue while achieving comparable image quality scores with a minimal model size.

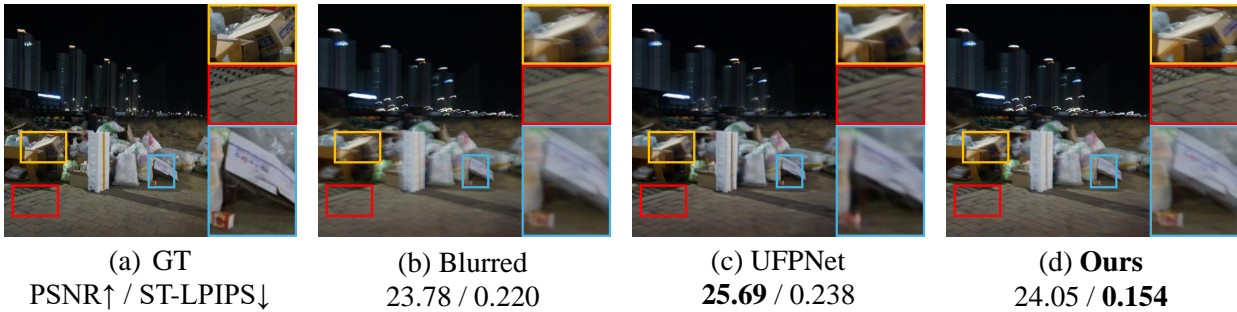

(a) GT
PSNR↑ / ST-LPIPS↓

(b) Blurred
23.78 / 0.220

(c) UFPNet
**25.69** / 0.238

(d) **Ours**
24.05 / **0.154**

Figure 2: Which deblurred image is clearer? The above example on RealBlur-J (Rim et al., 2020) suggests that ST-LPIPS (Ghildyal & Liu, 2022) effectively captures visual fidelity in real-world scenarios, whereas the distortion metric PSNR fails, emphasizing the importance of using perceptual metrics.

We next revisit the **limitations of distortion metrics** such as PSNR. As shown in Figure 2, our deblurred image deliver a lower PSNR but exhibits higher visual quality compared to UFPNet (Fang et al., 2023). Distortion metrics often fail to align with perceptual quality, which is why Learned Perceptual Image Patch Similarity (LPIPS) (Zhang et al., 2018) and the enhanced alternatives like Shift-tolerant LPIPS (ST-LPIPS) (Ghildyal & Liu, 2022) are gaining popularity for better correlating with human visual perception. In this paper, we employ ST-LPIPS since the real-world datasets contain image pairs with minor shifts. Figure 2 demonstrates that our method generates significantly sharper edges and clearer textures, with LPIPS and ST-LPIPS effectively capturing these perceptual improvements, whereas PSNR fails to reflect them.

To achieve **robust generalization capabilities** and **high perceptual performance**, we develop DeblurDiNAT, a Transformer architecture for real-world image deblurring. First, DeblurDiNAT is equipped with alternating local and global Transformer blocks, which are built upon dilated neighborhood attention (DiNA) (Hassani & Shi, 2022; Hassani et al., 2023) with small and large dilation factors. This structure with alternating dilation factors provides flexible receptive fields to capture blurred patterns with varying magnitudes.

To assist Transformer architectures to better understand cross-channel dependencies, we introduce a local cross-channel learner (LCCL) that only considers each channel's nearest neighbors. LCCL draws inspiration from channel attention (Hu et al., 2018; Wang et al., 2020). Building on this novelty, we propose a new self-attention framework, termed as channel-aware self-attention (CASA), where a parallel LCCL adaptively modulates the outputs of self-attention layers based on their shared inputs. Point-wise 2D convolutions can help capture global channel relationships, which is not the focus of this paper.

The other basic component of Transformers is the feed-forward networks (FFNs). Inspried by the simple gate without any non-linear activation in CNN-based NAFNet (Chen et al., 2022), we remove the GELU function (Hendrycks & Gimpel, 2016) from the gated-dconv feed-forward network (GDFN) (Zamir et al., 2022), and simply it to a linear divide and multiply feed-forward network (DMFN). It is worth mentioning that the non-linearity is ensured in other units of DeblurDiNAT, such as encoders and feature fusion modules.

In addition to the above novelties, we design a novel feature fusion paradigm for DeblurDiNAT. As no activation exists in our DMFN of decoders, non-linearity is introduced before the decoder path to effectively model complex visual data. To achieve this, we propose the non-linear lightweight dual-stage feature fusion (LDFF), an efficient method for merging and propagate visual features across the networks.

Our contributions are summarized as follows:

- We design a local and global blur learning strategy by alternating different dilation factors of DiNA, integrated with a local cross-channel learner to capture adjacent channel relationships.

- We present a simple while effective feed-forward network and a lightweight dual-stage feature fusion method for our Transformer architecture to effectively process complex blur information.

- Our proposed compact network, DeblurDiNAT, exhibits exceptional generalization capabilities and delivers high perceptual quality, while preserving a lightweight model design.

## 2 Related Work

**CNNs for Image Deblurring.** In the past decade, CNN-based methods have demonstrated significant effectiveness in image deblurring tasks (Hradiš et al., 2015; Svoboda et al., 2016; Nah et al., 2017; Kupyn et al., 2019; Tao et al., 2018; Gao et al., 2019; Zhang et al., 2019; Cho et al., 2021; Zamir et al., 2021; Chen et al., 2022; Fang et al., 2023). MIMO-UNet (Cho et al., 2021) introduces a concatenate-and-convolve strategy to combine multi-scale and same-scale features within a U-Net architecture. MPRNet (Zamir et al., 2021) innovatively proposes a multi-stage progressive pipeline that significantly enhances deblurring performance. NAFNet (Chen et al., 2022) presents a simple yet effective baseline for high-quality image restoration, while UFPNet (Fang et al., 2023) further improves deblurring performance by incorporating kernel estimation. To capture global blurred patterns, the latest CNN-based methods, including NAFNet and UFPNet (Chen et al., 2022; Fang et al., 2023), have been scaled to very large sizes to achieve a large receptive field.

**Transformers for Image Deblurring.** More recently, Transformer networks have been studies for the image deblurring problem (Wang et al., 2022; Tsai et al., 2022; Zamir et al., 2022; Kong et al., 2023; Mao et al., 2024). Restormer (Zamir et al., 2022) computes cross-covariance across feature channels to model global connectivity. FFTformer (Kong et al., 2023), a novel frequency-based Transformer, achieves state-of-the-art (SOTA) performance on the known domains. The latest LoFormer (Mao et al., 2024) incorporates channel-wise self-attention in the frequency domain and achieves deblurred results comparable to FFTformer. However, these Transformer-based approaches lack exploration on model generalization to unseen real-world domains, and they do not adequately quantify perceptual performance either.

## 3 Method

**Preliminaries.** We adopt Dilated Neighborhood Attention (DiNA) (Hassani & Shi, 2022; Hassani et al., 2023) for image deblurring. DiNA serves as a flexible SA mechanism for short- and long-range learning by adjusting the dilation factor without additional complexity theoretically. For simplicity, consider a feature

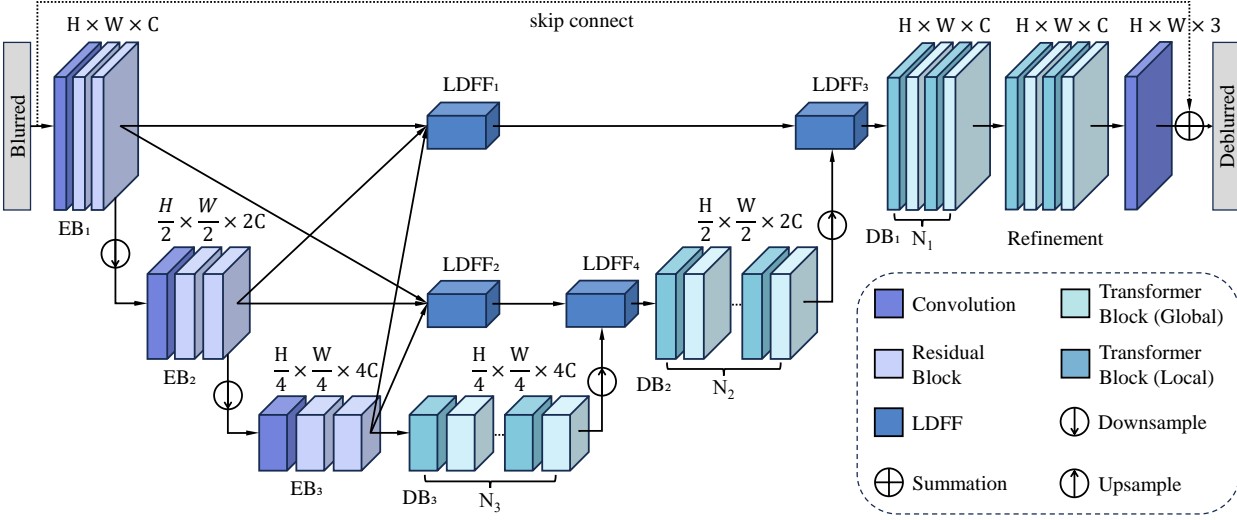

Figure 3: Architecture of DeblurDiNAT for single image deblurring. DeblurDiNAT is a hybrid encoder-decoder Transformer, where cascaded residual blocks extract multi-scale image features from input images, and the alternating local and global Transformer blocks reconstructs clean images from hierarchical features.

map $\mathbf{X} \in \mathbb{R}^{n \times d}$, where $n$ is the number of tokens (basic data units) and $d$ is the dimension; the *query* and *key* linear projections of $\mathbf{X}$, $Q$ and $K$; and relative positional biases between two tokens $i$ and $j$, $B(i,j)$. Given a dilation factor $\delta$ and neighborhood size $k$, the attention weights for the $i$-th token is defined as:

$$\mathbf{A}_i^{(k,\delta)} = \begin{bmatrix} Q_i K_{\rho_1^\delta(i)}^T + B_{(i,\rho_1^\delta(i))} \\ Q_i K_{\rho_2^\delta(i)}^T + B_{(i,\rho_2^\delta(i))} \\ \vdots \\ Q_i K_{\rho_k^\delta(i)}^T + B_{(i,\rho_k^\delta(i))} \end{bmatrix}. \tag{1}$$

The corresponding matrix $\mathbf{V}_i^{(k,\delta)}$, whose elements are the $i$-th token's $k$ adjacent *value* linear projections, can be obtained by:

$$\mathbf{V}_i^{(k,\delta)} = \begin{bmatrix} V_{\rho_1^\delta(i)}^T & V_{\rho_2^\delta(i)}^T & \cdots & V_{\rho_k^\delta(i)}^T \end{bmatrix}^T. \tag{2}$$

The output feature map of DiNA for the $i$-th token is achieved by:

$$\mathrm{DiNA}_k^\delta(i) = softmax\left(\frac{\mathbf{A}_i^{(k,\delta)}}{\sqrt{d_k}}\right)\mathbf{V}_i^{(k,\delta)}. \tag{3}$$

### 3.1 Alternating Dilation Factor Structure

As shown in Figure 3 and 4 (b.1) and (b.2), to capture both local and global blurred patterns, the decoder path of DeblurDiNAT alternates the Transformer blocks containing DiNA with a dilation factor of $\delta \in \left\{1, \lfloor \frac{n}{k} \rfloor\right\}$, where $n$ is the feature size, and $k$ is the kernel size set to a constant number 7 in our full model. The self-attention window size is widened by incrementing the value of dilation factor, enabling the network to efficiently learn larger-area blurs.

### 3.2 Channel Aware Self Attention

Figure 4 (a) shows that channel aware self-attention (CASA) contains two parallel units, DiNA and a local cross-channel learner (LCCL). As illustrated in Figure 4 (c), an LCCL first transforms the normalized 2-D features into 1-D data by global average pooling (GAP); then, it applies a 1-D convolution to the intermediate features along the channel dimension; finally, a *sigmoid* function is adopted to compute attention scores. The outputs of the LCCL and DiNA are merged by element-wise multiplications.

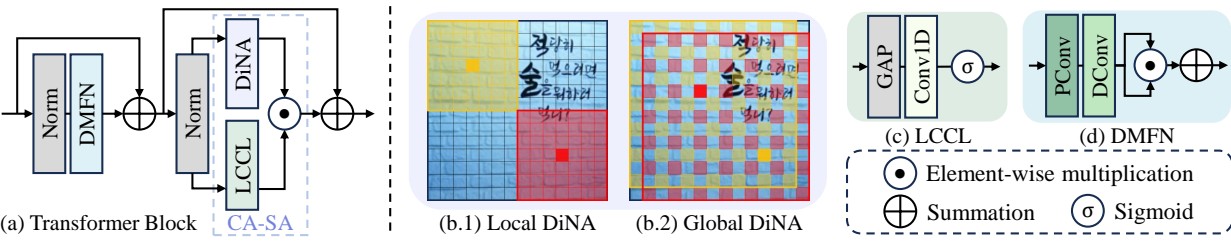

Figure 4: Structures of (a) Transformer block, (b.1) local DiNA, (b.2) global DiNA, (c) local cross-channel learner (LCCL), and (d) divide-and-multiply feed-forward network (DMFN) in DeblurDiNAT. Note: GAP represents global average pooling; PConv and Dconv refer to point-wise and depth-wise convolutions; Conv1D denotes the convolution applied to 1D sequences. Tensor transformations are omitted for simplicity.

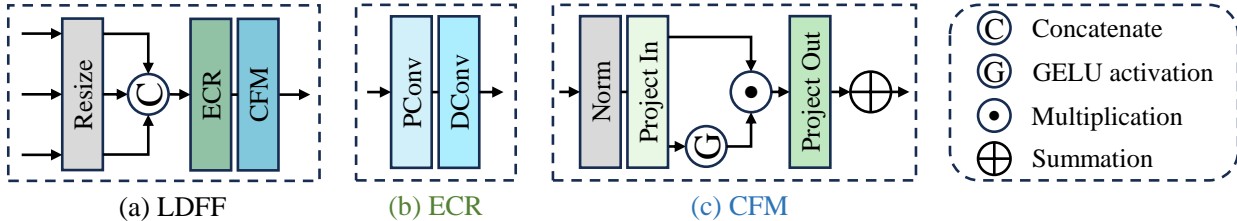

Figure 5: Structures of (b) lightweight dual-stage feature fusion (LDFF), composed of (b.1) efficient channel reduction (ECR) and (b.2) complementary feature mixer (CFM).

Given a layer normalized input tensor $\mathbf{X} \in \mathbb{R}^{\hat{H} \times \hat{W} \times \hat{C}}$. The output of CASA is computed as,

$$\hat{\mathbf{X}} = \mathrm{DiNA}(\mathbf{X}) \odot \mathrm{LCCL}(\mathbf{X}),$$
$$\mathrm{LCCL}(\mathbf{X}) = f^{-1}(\mathrm{Conv}_{1d}(f(\mathrm{GAP}_{2d}^{1 \times 1}(\mathbf{X})))), \tag{4}$$

where $\odot$ denotes element-wise multiplication; $f$ is a tensor manipulation function which squeezes and transposes a $C \times 1 \times 1$ matrix, resulting in a $1 \times C$ matrix; $\mathrm{Conv}_{1d}$ denotes a 1D convolution with a kernel size of 3; $\mathrm{GAP}_{2d}^{1 \times 1}$ indicates global average pooling, outputting a tensor of size $1 \times 1$.

### 3.3 Divide and Multiply Feed-Forward Network

As shown in Figure 4 (d), the divide and multiply feed-forward network (DMFN) omits the activation function in previous Transformer FFNs, inspired by the simple gate in CNN-based NAFNet (Chen et al., 2022). We assert that DeblurDiNAT remains a robust model because the Leaky ReLU in the encoder path ensures non-linearity in latent features, and the feature fusion modules with GELU propagate non-linear features cross different network levels. From a layer normalized input feature $\mathbf{X} \in \mathbb{R}^{\hat{H} \times \hat{W} \times \hat{C}}$, DMFN first applies point-wise convolutions to expand the feature channels by a factor $\gamma = 2$. After that, $3 \times 3$ depth-wise convolutions are utilized to aggregate spatial context, generating $\mathbf{X}_0 \in \mathbb{R}^{\hat{H} \times \hat{W} \times 2\hat{C}}$. Next, the intermediate feature map $\mathbf{X}_0$ is equally divided along the channel dimension, yielding $\mathbf{X}_1$ and $\mathbf{X}_2$, so that their element-wise multiplication result is of shape $\mathbb{R}^{\hat{H} \times \hat{W} \times \hat{C}}$. The overall process of DMFN is formulated as:

$$\hat{\mathbf{X}} = W_d^1 W_p^1 \mathbf{X} \odot W_d^2 W_p^2 \mathbf{X}, \tag{5}$$

where $W_p^{(\cdot)}$ and $W_d^{(\cdot)}$ denote the convolution with filter size of $1 \times 1$ and $3 \times 3$.

### 3.4 Lightweight Dual-Stage Feature Fusion

As shown in Figure 5 (a)-(c), LDFF first resizes input features from different encoder levels, and concatenates them along the channel dimension. LDFF is composed of two stages, efficient channel reduction (ECR) and complementary feature mixing (CFM). In ECR, a $1 \times 1$ convolution is followed by a $3 \times 3$ depth-wise convolution. Then, CFM splits a normalized feature map into two complementary features by convolutional

layers, passing through two parallel branches, one of which is gated by GELU activation. Finally, the two complementary features are merged by element-wise multiplications and additional convolutions. The two multi-scale LDFF blocks in DeblurDiNAT are formulated as:

$$
\begin{aligned}
\text{LDFF}_1^{out} &= \text{LDFF}_1(\text{EB}_1^{out}, (\text{EB}_2^{out})^{\times 2}, (\text{EB}_3^{out})^{\times 4}), \\
\text{LDFF}_2^{out} &= \text{LDFF}_2((\text{EB}_1^{out})^{\times \frac{1}{2}}, \text{EB}_2^{out}, (\text{EB}_3^{out})^{\times 2}),
\end{aligned}
\tag{6}
$$

where $\text{LDFF}_i^{out}$ and $\text{EB}_i^{out}$ denote the output features of the $i$-th LDFF and the $i$-th encoder block; $\times 2$ and $\times 4$ represent upscaling by 2 and 4; $\times \frac{1}{2}$ means downscaling by 2. LDFF for same-scale feature fusion is of a similar architecture without the resizing procedure.

## 4 Experiments and Analysis

**Datasets.** The GoPro dataset (Nah et al., 2017) contains a variety of outdoor scenes, while the HIDE dataset (Shen et al., 2019) is distinct for its focus on human images. Both datasets share similar data collection methods. RealBlur-R (Rim et al., 2020) and RealBlur-J (Rim et al., 2020) are closer to real-world blurred images, with most images captured in low-light conditions. RWBI (Zhang et al., 2020) is another real-world dataset but does not include ground-truth images. Since the focus of this paper is generalization capabilities and perceptual performance, DeblurDiNAT was trained only on the GoPro training set and was directly applied to HIDE, RealBlur-R, RealBlur-J, and RWBI test sets.

**Implementation Details.** We propose DeblurDiNAT-S (Small) and -L (Large). The number of Transformer blocks in DeblurDiNAT-S are $[4, 6, 8]$, attention heads are $[2, 4, 8]$, and the number of channels are $[64, 128, 256]$ at each level. In the large network, one more residual block is appended to each encoder; the number of Transformer blocks increment to $[6, 12, 18]$. We trained both models using Adam optimizer with the initial learning rate of $2 \times 10^{-4}$ which was gradually decayed to $10^{-7}$ by the cosine annealing strategy. The batch size was set to 8. DeblurDiNAT-L converged after training for 4000 epochs with a patch size of $256 \times 256$ and another 2000 epochs with a patch size of $496 \times 496$, on eight NVIDIA A100 GPUs.

**Evaluation Metrics.** Following previous work (Whang et al., 2022), FID↓ (Fréchet Inception Distance) (Heusel et al., 2017), KID↓ (Kernel Inception Distance) (Bińkowski et al., 2018), LPIPS↓ (Zhang et al., 2019) and NIQE↓ (Mittal et al., 2012) are employed as perceptual metrics. We adopt the Shift-Tolerant Perceptual Similarity Metric (ST-LPIPS↓, **abbreviated as S-LP in Tables**) (Ghildyal & Liu, 2022) as an alternative to LPIPS (Zhang et al., 2018). FID and KID are normalized to a range of $[0, 1]$. We also evaluate image quality by distortion metrics such as PSNR↑ and SSIM↑ for a fair comparison.

### 4.1 Comparisons with State-of-the-art Methods

In this paper, the deblurring networks with parameter count larger than 40M are considered as **non-compact networks**, labeled with $^\dagger$ in Tables 1 and 2. Since the focus of this paper is generalization and visual fidelity, we evaluate the performance of GoPro-trained models on unknown domains, and provide the perceptual metric results in Table 1 and deblurred images in Figure. We also report distortion metrics in Table 2.

#### 4.1.1 Quantitative Comparisons

Tables 1 demonstrates that our proposed DeblurDiNAT-L achieves comparable or superior perceptual performance to state-of-the-art methods across multiple metrics while utilizing favorably fewer parameters. Specifically, compared to the other compact networks MIMO-Unet+ (Cho et al., 2021), MPRNet (Zamir et al., 2021), Restormer (Zamir et al., 2022) and FFTformer (Kong et al., 2023), our DeblurDiNAT-L achieves **significantly improved deblurring performance**. Compared to the closest competitor, UFPNet (Fang et al., 2023), DeblurDiNAT-L delivers on-par performance with **only 20% of the parameters**. While Figure 2 has implied that PSNR is not a good option to evaluate perceptual performance, we report distortion metrics in Table 2. Compared to recent deblurring Transformers such as FFTformer (Kong et al., 2023) and Lo-Former (Mao et al., 2024), Table 2 reports that DeblurDiNAT-L exhibits competitive distortion metric scores on real-world datasets. A detailed analysis of quantitative results on each dataset is provided as follows.

Table 1: Perceptual metric scores of GoPro-trained (Nah et al., 2017) models on HIDE (Shen et al., 2019), RealBlur-J and RealBlur-R (Rim et al., 2020) test sets. Best results are **highlighted**, the second best results are underlined, and the third highest results are double-underlined. In this paper, the deblurring networks with parameter count more than 40M are considered as non-compact models marked with †.

| Models | #P. (M) | HIDE | | | | RealBlur-J | | | | RealBlur-R | | | |
|---|---|---|---|---|---|---|---|---|---|---|---|---|---|
| | | FID | KID | S-LP | NIQE | FID | KID | S-LP | NIQE | FID | KID | S-LP | NIQE |
| MIMO-Unet+ 2021 | 16.1 | 0.178 | 0.062 | 0.00 | 3.24 | 0.965 | 0.770 | 0.142 | 3.99 | 0.637 | 0.273 | 0.034 | 5.34 |
| MPRNet 2021 | 20.1 | 0.168 | 0.054 | 0.070 | 3.46 | 0.606 | 0.333 | 0.103 | 3.97 | 0.550 | 0.243 | 0.031 | 5.44 |
| NAFNet† 2022 | 67.9 | 0.137 | 0.043 | 0.058 | 3.22 | 0.699 | 0.435 | 0.114 | **3.77** | 0.467 | 0.133 | 0.030 | 5.36 |
| UFPNet† 2023 | 80.3 | 0.150 | 0.065 | **0.051** | **3.13** | **0.460** | **0.255** | **0.081** | 4.07 | **0.402** | 0.135 | 0.025 | **5.17** |
| Restormer 2022 | 26.1 | 0.150 | 0.052 | 0.063 | 3.42 | 0.576 | 0.318 | 0.099 | 3.92 | 0.436 | 0.166 | 0.026 | 5.23 |
| Uformer-B† 2022 | 50.9 | 0.141 | 0.052 | 0.065 | 3.40 | 0.515 | 0.261 | 0.088 | 3.86 | 0.421 | 0.129 | **0.024** | 5.28 |
| FFTformer 2023 | 16.6 | 0.139 | 0.051 | 0.054 | 3.29 | 0.907 | 0.721 | 0.128 | 3.90 | 0.500 | 0.157 | 0.027 | 5.24 |
| LoFormer-L† 2024 | 49.0 | 0.125 | 0.040 | 0.052 | 3.30 | 0.672 | 0.416 | 0.110 | 3.79 | 0.454 | 0.157 | 0.027 | **5.17** |
| **DeblurDiNAT-S** | **9.1** | 0.143 | 0.045 | 0.063 | 3.43 | 0.578 | 0.315 | 0.095 | 3.83 | 0.458 | 0.122 | 0.027 | 5.21 |
| **DeblurDiNAT-L** | 16.1 | **0.121** | **0.034** | 0.053 | 3.49 | 0.509 | 0.265 | 0.084 | 3.79 | 0.417 | **0.104** | 0.025 | **5.17** |

Table 2: Distortion metric comparisons on HIDE (Shen et al., 2019) and RealBlur (Rim et al., 2020) datasets. ⋆ means that we directly use the image quality values reported in the original paper (Fang et al., 2023).

| Models | Param. (M) | Average PSNR | HIDE | | RealBlur-J | | RealBlur-R | |
|---|---|---|---|---|---|---|---|---|
| | | | PSNR | SSIM | PSNR | SSIM | PSNR | SSIM |
| MIMO-Unet+ | 16.1 | 31.05 | 29.99 | 0.930 | 27.63 | 0.837 | 35.54 | 0.947 |
| MPRNet | 20.1 | 31.88 | 30.96 | 0.939 | 28.70 | 0.873 | 35.99 | 0.952 |
| NAFNet† | 67.9 | 31.87 | 31.32 | 0.943 | 28.32 | 0.857 | 35.97 | 0.951 |
| UFPNet†⋆ | 80.3 | **32.62** | **31.74** | 0.947 | **29.87** | 0.884 | **36.25** | 0.947 |
| Restormer | 26.1 | 32.12 | 31.22 | 0.942 | 28.96 | 0.879 | 36.19 | **0.957** |
| Uformer-B† | 50.9 | 32.06 | 30.90 | **0.953** | 29.09 | **0.886** | 36.19 | 0.956 |
| FFTformer | 16.6 | 31.75 | 31.62 | 0.946 | 27.75 | 0.853 | 35.87 | 0.953 |
| LoFormer-L† | 49.0 | 32.16 | 31.86 | 0.949 | 28.53 | 0.863 | 36.09 | 0.955 |
| **DeblurDiNAT-S** | **9.1** | 31.86 | 30.92 | 0.939 | 28.73 | 0.874 | 35.93 | 0.953 |
| **DeblurDiNAT-L** | 16.1 | 32.18 | 31.47 | 0.944 | 28.98 | 0.885 | 36.09 | 0.955 |

**HIDE.** The fine facial textures in the human centric images elevate NIQE values (Zvezdakova et al., 2019). We still report NIQE on the human centric dataset HIDE (Shen et al., 2019) for thorough comparisons. As shown in Table 1, the proposed method achieves the state-of-the-art FID and KID, while delivering comparable ST-LPIPS to UFPNet (Fang et al., 2023). It proves that our model demonstrates great visual performance and generalizes very well to unknown images with human beings.

**RealBlur-J.** Table 1 reports that our DeblurDiNAT-L ranks right after UFPNet (Fang et al., 2023) in terms of FID and ST-LPIPS while with 79.95% fewer parameters. Notably, our method overperforms UFPNet (Fang et al., 2023) by 0.28 as rated by NIQE. It verifies that the proposed approach restores real-world blurred images favorably close to human perception.

**RealBlur-R.** Table 1 shows that DeblurDiNAT-L delivers the state-of-the-art KID and NIQE, followed by our small version DeblurDiNAT-S. Compared to UFPNet (Fang et al., 2023), DeblurDiNAT-S achieves competitive KID, ST-LPIPS and NIQE while with only 11.33% parameters on RealBlur-R (Rim et al., 2020). It suggests the superiority of our method in handling real-world blurs with low-light illuminations.

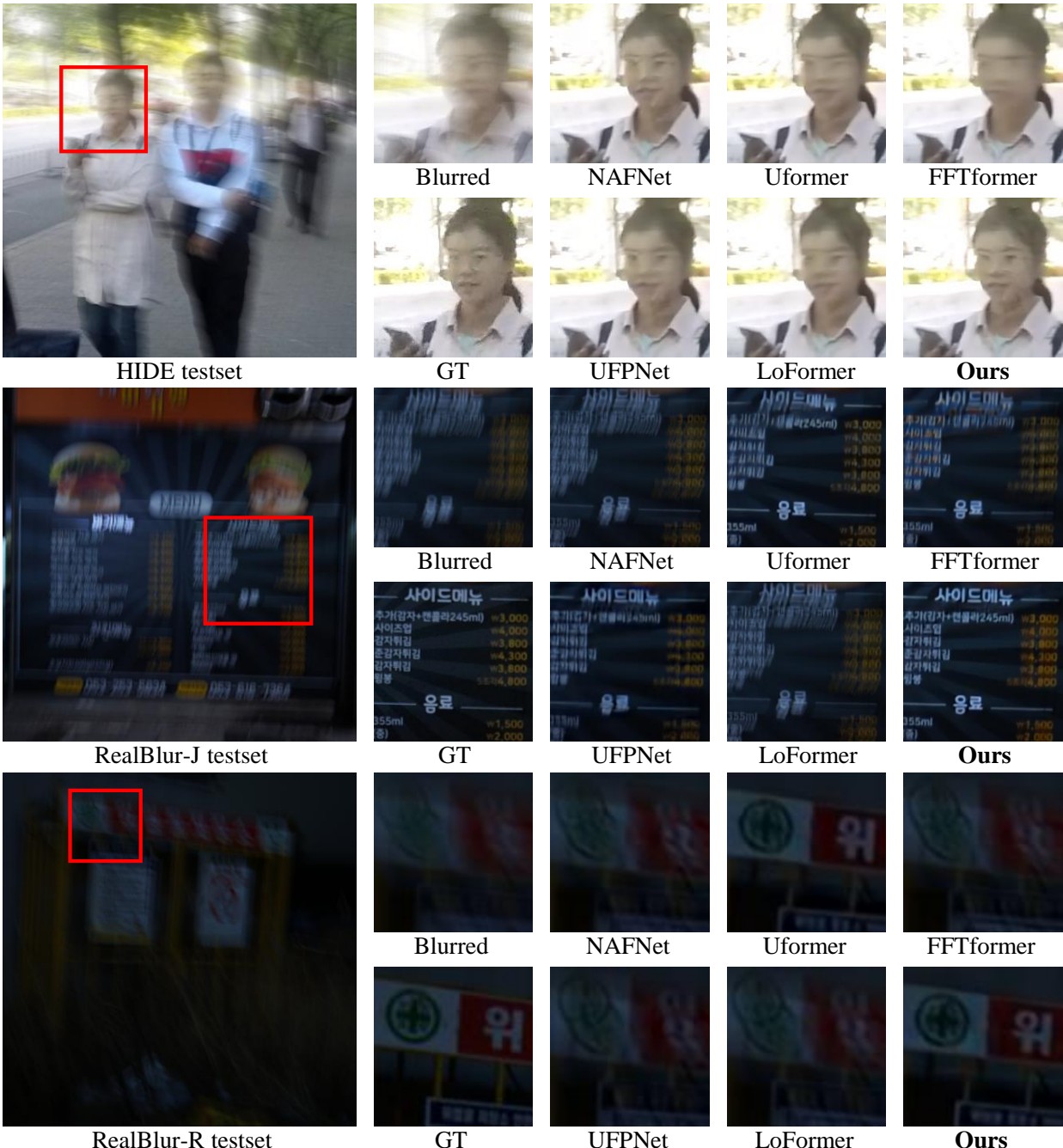

Figure 6: Deblurred images of GoPro-trained models on the unknown HIDE (Shen et al., 2019), RealBlur-J and -R (Rim et al., 2020) datasets. Best viewed with zoom and high brightness for details.

### 4.1.2 Qualitative Comparisons

As illustrated in Figure 6, our proposed model, DeblurDiNAT-L, outperforms recent deblurring networks by generating noticeably clearer human faces on the HIDE dataset (Shen et al., 2019). On real-world datasets such as RealBlur-J and RealBlur-R (Rim et al., 2020), DeblurDiNAT-L demonstrates superior performance by producing cleaner and sharper text characters, particularly under challenging low-light conditions. These results highlight the generalization capabilities and visual fidelity of our model compared to existing approaches. Notably, this performance is achieved with a minimal number of parameters.

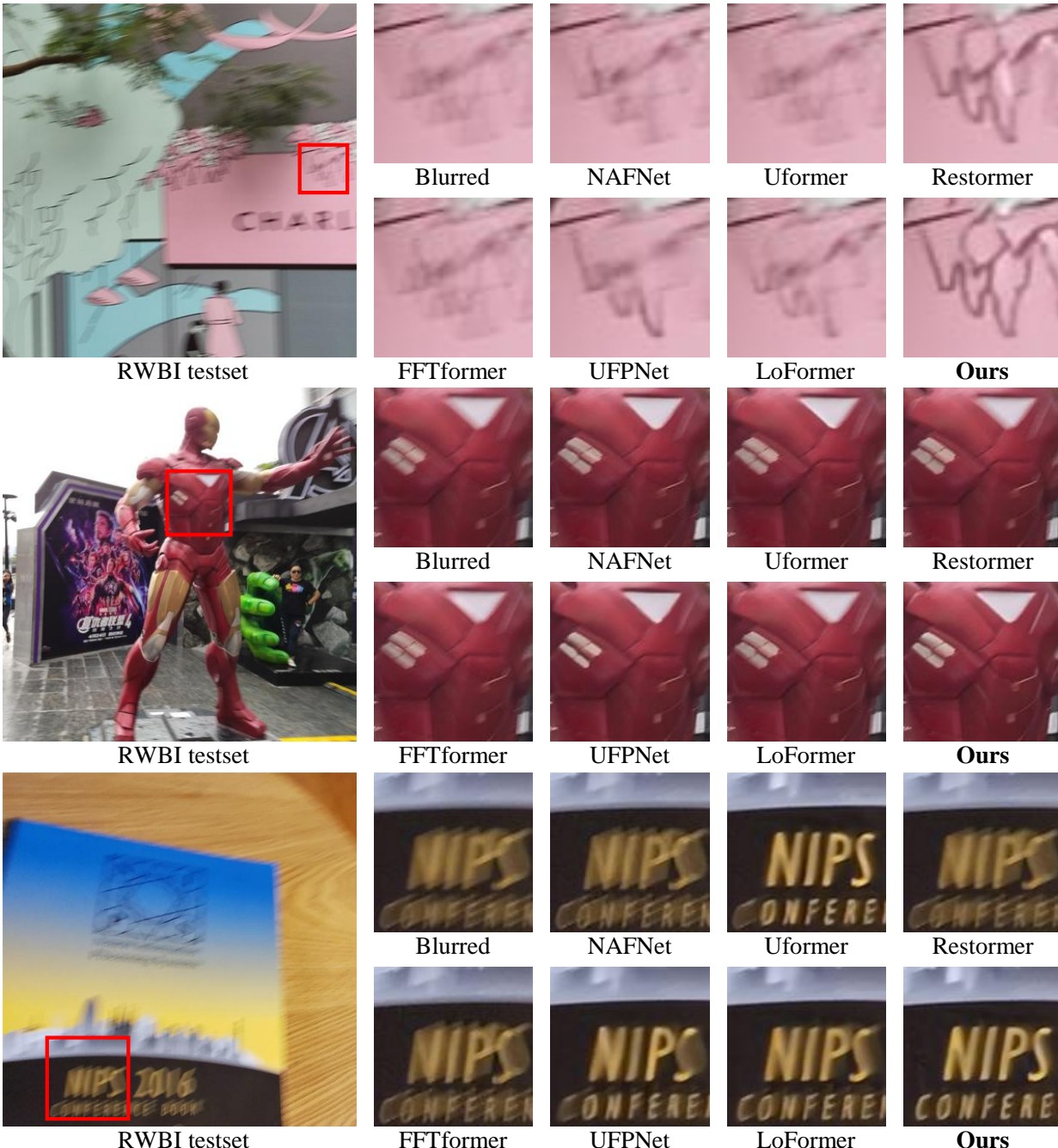

Figure 7: Deblurred images of GoPro-trained models on the unknown RWBI dataset (Zhang et al., 2020).

**RWBI.** Figure 7 demonstrates that the proposed DeblurDiNAT-L effectively restores significantly sharper streaks compared to existing methods on RWBI (Zhang et al., 2020). For instance, in the middle row (iron man) of Figure 7, the forward-slash-shaped line at the top right is restored with remarkable clarity.

## 4.2 Ablation Study

We conduct ablation experiments to evaluate each component of DeblurDiNAT. Models were trained on the GoPro dataset (Nah et al., 2017) with a patch size of $128 \times 128$ for 3000 epochs only. Given that

Table 3: Effects of different feature fusion strategies, AFF (Cho et al., 2021) and our LDFF. Exploration on combinations of different dilation factors, the lower bound only, the upper bound only and the hybrid.

| Feature Fusion | | Dilation Factor | | HIDE | | RealBlur-J | |
|---|---|---|---|---|---|---|---|
| AFF | **LDFF** | $\delta = 1$ | $\delta = \lfloor \frac{n}{k} \rfloor$ | S-LP↓ | PSNR↑ | S-LP↓ | PSNR↑ |
| ✓ | | | ✓ | 0.084 | 29.76 | **0.097** | 28.51 |
| | ✓ | | ✓ | 0.082 | 29.86 | 0.098 | **28.55** |
| | ✓ | ✓ | | 0.085 | 29.47 | 0.101 | 28.34 |
| | ✓ | ✓ | ✓ | **0.078** | **29.94** | 0.098 | 28.54 |

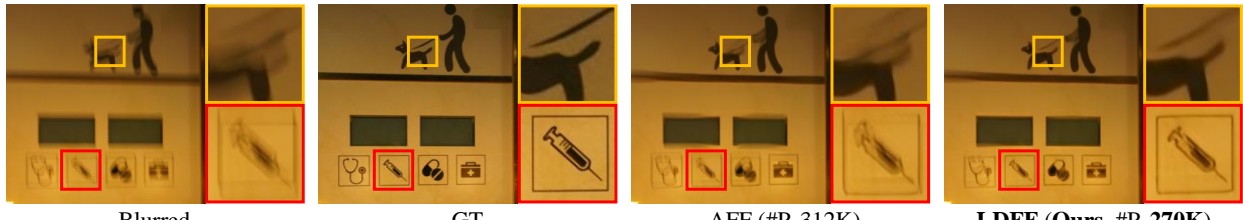

Figure 8: The proposed feature fusion method improves visual quality with 13.46% smaller module size.

tradeoff between reconstruction accuracy and perceptual quality exists (Blau et al., 2018), both quantitative comparisons and qualitative results are provided to thoroughly demonstrate the effectiveness of our methods.

**Feature Fusion.** We first compare the proposed LDFF with the previous AFF in MIMO-Unet (Cho et al., 2021). As shown in Table 3, LDFF improves both perceptual and distortion metrics on HIDE, and achieves similar ST-LPIPS and noticeably higher PSNR on RealBlur-J. Figure 8 shows that our method produces sharper edges, verifying its superior utilization of multi-scale features. Notably, the lightweight dual-stage feature fusion (LDFF) in DeblurDiNAT contains only **270K** parameters, significantly fewer than the 312K parameters required by the feature fusion methods in MIMO-Unet (Cho et al., 2021).

**Dilation Factor.** We experiment with different combinations of dilation factors, $\delta = 1$, $\delta = \lfloor \frac{n}{k} \rfloor$, and the hybrid $\delta \in \left\{ 1, \lfloor \frac{n}{k} \rfloor \right\}$. As shown in Table 3, the presented structure incorporating both small and big dilation factor obtains the best overall performance. Furthermore, Figure 9 suggests that the introduction of small dilation factors effectively reduces artifacts in fine areas, resulting in smoother restorations.

**Channel Aware Self Attention.** We evaluate the effectiveness of CA-SA by comparing it with self-attention that excludes the Local Cross-Channel Learner (LCCL). As presented in Table 4, CA-SA achieves consistent quantitative improvements across various metrics on multiple datasets. Additionally, the hue analysis in Table 4 reveals that CA-SA produces colors much closer to the reference, highlighting the LCCL module's ability to enhance the network's cross-channel learning, particularly for color information. Figure 10 illustrates that CA-SA noticeably improves the visual quality of deblurred images.

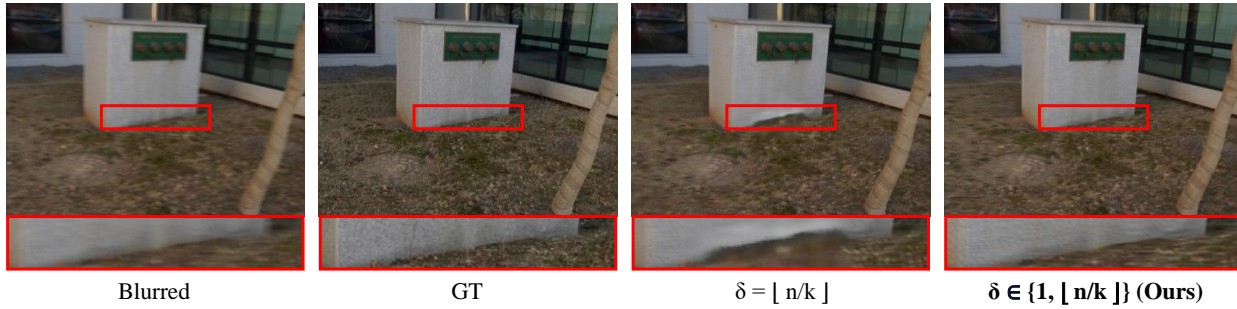

Figure 9: The presented alternating structure with hybrid dilation factors reduces artifacts in small regions.

Table 4: Investigation on the proposed channel-aware self-attention (CA-SA) with a local cross channel learner (LCCL), and comparison between GDFN (Zamir et al., 2022) and the presented DMFN. Hue Distance denotes the average difference between the hue of deblurred images and ground truth in the HSV color space.

| Self-Attention | | FFN | | HIDE | | RealBlur-J | | Hue Distance |
| --- | --- | --- | --- | --- | --- | --- | --- | --- |
| SA | **CA-SA** | GDFN | **DMFN** | S-LP↓ | PSNR↑ | S-LP↓ | PSNR↑ | % |
| ✓ | | ✓ | | 0.078 | 29.94 | 0.098 | 28.54 | 0.65 |
| | ✓ | ✓ | | 0.080 | 29.97 | 0.096 | 28.61 | **0.58** |
| ✓ | | | ✓ | 0.079 | 29.93 | 0.096 | 28.53 | 0.70 |
| | ✓ | | ✓ | **0.077** | **29.99** | **0.096** | **28.59** | **0.58** |

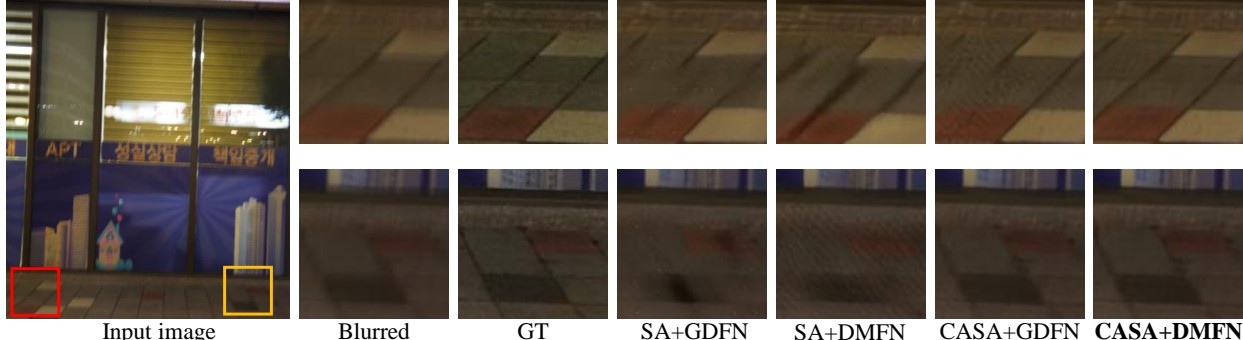

Input image     Blurred     GT     SA+GDFN     SA+DMFN     CASA+GDFN     **CASA+DMFN**

Figure 10: The proposed methods CA-SA and DMFN generate road textures closer to the ground truth.

**Feed-Forward Network.** We compare the proposed Divide-and-Multiply Feed-Forward Network (DMFN) with the Gated FFN (GDFN) (Zamir et al., 2022) in DeblurDiNAT. Since non-linearity has been ensured in the encoder path and feature fusion modules, DeblurDiNAT eliminates the need for GELU activation in FFN. As shown in Table 4, DMFN delivers competitive performance compared to GDFN, showcasing its efficiency. Figure 10 reveals that our method effectively captures boundaries between distinct regions.

## 5 Conclusion

In this work, we propose DeblurDiNAT, a novel deblurring Transformer that combines strong generalization capabilities and impressive visual fidelity with a minimal model size. First, an alternating structure with different dilation factors is introduced to effectively estimate blurred patterns of varying magnitudes. Second, a local cross-channel learner adaptively modulates self-attention outputs, enhancing channel-wise learning for attributes such as color information. To enable efficient feature propagation, a simple yet effective feed-forward network is employed, replacing traditional Transformer designs. Additionally, a lightweight dual-stage feature fusion method ensures non-linearity and aggregates multi-scale features for improved performance. Comprehensive experiments demonstrate that DeblurDiNAT achieves comparable quantitative results and significantly superior qualitative performance compared to the state-of-the-art UFPNet (Fang et al., 2023), while utilizing only 20% of the parameters.

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
