# OpenReview forum: "DeblurDiNAT: A Compact Model with Exceptional Generalization and Visual Fidelity on Unseen Domains"
_TMLR — Withdrawn by Authors_

### Review · Reviewer_pecR · 2025-03-26

**Summary Of Contributions:**

Focusing on image deblurring, this paper presents a modified transformer architecture that consists of:
1. Alternating local and global transformer blocks, implemented via dilated neighborhood attention
2. A form of cross-channel attention
3. A feed-forward without non-linearity
4. But then, a non-linearity is introduced before the decoder.

Experiments demonstrate that the proposed method performs on par (and even outperforms) existing architectures of the same number of parameters, but larger models (especially UFPNet) tend to outperform DeblurDiNAT-L.

**Audience:**

Yes

**Claims And Evidence:**

Yes

**Requested Changes:**

I believe that if the key argument of the authors is improved performance per model size, they should demonstrate this. It is absolutely the case for a small model size. I would make the same comparison, but for #Parameters of 70-80 M. Otherwise, the authors should better clarify what they mean by "limited generalization problem."

Eq. (1) -- \rho is not defined. Please fix.

In general, there are MANY acronyms in the paper, e.g., LPIPS, ST-LPIPS, DiNA, LCCL, CASA, GDFN, DMFN, and the list goes on and on. This makes the paper difficult to read and follow.

**Strengths And Weaknesses:**

Disclaimer: This work is mainly experimental, and I do not consider myself an expert in this particular line of research.

Strengths:
1. I found the idea of using local and global transformer blocks and cross-channel attention natural for capturing different/varying blurring patterns.
2. The experiments are extensive, and the presented ablation study demonstrates the gain of each component.

Weaknesses:
1. The authors argue that their method aims to tackle the "limited generalization problem," but from the experiments, it seems that UFPNet performs better than DeblurDiNAT. I understand that there are failure cases, but do they happen frequently?
2. Related to the above, at some point in the paper---and somewhat out of the blue---the authors have started to make a distinction between small and large models, arguing that the proposed method has better performance compared to the model size. This is a valid point, but it was never highlighted before as one of the goals of the work. Further, it seems that UFPNet (which is not a huge model) performs quite well (in all metrics).

---

> ### Author Response · Authors · 2025-03-28
> **Response to Reviewer pecR of Paper3985**
>
> We greatly appreciate the reviewer’s comments! Before submitting our revisions, we would like to address your concerns.
>
> Response to Weakness 1:
>
> We acknowledge the reviewer's observation that UFPNet (Fang et al., 2023) outperforms our DeblurDiNAT on certain metrics and datasets as shown in Table 1 and 2.
>
> However, in real-world scenarios where no ground-truth reference is available, such as the RWBI dataset (Zhang et al., 2020), robustness is critical. Notably, UFPNet (Fang et al., 2023) exhibits 18 failure cases out of 1000 images, whereas our method has demonstrated no observable failures. Given that image deblurring is widely deployed on cameras and mobile devices, a ~2% failure rate is significant and poses a practical concern.
>
> Furthermore, as illustrated in Figure 1, 6 and 7, our model (16.1M parameters) produces visibly cleaner restorations compared to UFPNet (80.3M parameters), which frequently struggles with textures such as windows and roads. This highlights the efficiency and reliability of our approach in real-world applications.
>
> Response to Weakness 2:
>
> As shown in Table 1, UFPNet (Fang et al., 2023)  is an ultra-large model among recent image deblurring networks, with 80.3M parameters, whereas our DeblurDiNAT-L contains only 16.1M parameters. According to the original UFPNet paper, its main contribution is a kernel estimation module designed to enhance existing deblurring networks, such as NAFNet (Chen et al., 2022). However, UFPNet is built upon the already large NAFNet, increasing its size from 67.9M to 80.3M parameters, while also doubling the training iterations compared to NAFNet.
>
> We attribute UFPNet’s failures in real-world scenarios to overfitting caused by its intensive training strategy. While this approach may improve benchmark performance, it appears to hinder generalization to unseen real-world cases, as reflected in its higher failure rate on the RWBI dataset.
>
> One challenge we face in conducting extensive experiments with UFPNet to analyze performance versus model size is that the current state-of-the-art UFPNet only provides test codes. Additionally, their exact training strategy and training dataset format are not publicly available.
>
> We appreciate your feedback and look forward to your response. Respectfully, the authors.

---

### Review · Reviewer_vkx1 · 2025-03-30

**Summary Of Contributions:**

The paper introduces a model that shows better deblurring performance compared to previous methods.

**Audience:**

Yes

**Broader Impact Concerns:**

The paper works on deblurring algorithm. No direct broader ethical impact concerns exist.

**Claims And Evidence:**

No

**Requested Changes:**

- Introduction contains results instead of providing extensive background of the results. Paragraphs 3 and 4 can be moved to results section and the space can be used for more extensive dive into literature or a shorter introduction.
Possible extensions:
- How the human brain achieves good improvement of degraded images that is generalizable (https://www.eneuro.org/content/5/3/ENEURO.0443-17.2018.short; https://pubmed.ncbi.nlm.nih.gov/22841311/; https://opg.optica.org/josaa/abstract.cfm?uri=JOSAA-20-7-1434; https://www.jneurosci.org/content/37/28/6638.abstract; https://journals.physiology.org/doi/full/10.1152/jn.00812.2009) and possible mechanisms that can be inspired from it (https://elifesciences.org/articles/38105; https://www.biorxiv.org/content/10.1101/2022.03.07.483196v1.abstract; https://openreview.net/forum?id=rylU4mtUIS)
- More deep dive into deblurring algorithms (check this list: https://github.com/subeeshvasu/Awesome-Deblurring)

- Explanation of dual-fusion part in abstract mentions an alternative approach without explaining what approach that is.
- Many acronyms are used without or before introduction PSNR, SSIM, GT, LDFF, EB. There are also cases where the use of acronym is not very useful such as GT in the figures despite having ample white space to write the full word + the full wording is not too lengthy overall.

- Figure 2 is really hard to see the improvement given the size. I had to zoom in to like 250% to see it. I suggest removing the complete image and increase the size of the patches. Or at least show the big picture only once.

- Mode collapse is not explained or reasoning discussed

Methods section needs a massive reorganization:
- Figure 3 needs to be explained on its own first to explain the overall architecture. I suggest also changing the color labelling to a more diverse hues because the differences between the blues are hard to distinguish
- I suggest removing the section preliminaries and referring to the pieces explained there in the part where each part explained in the later subsections.
- The sub-systems are explained in an order that does not seem to follow the order of processing where LDFF for example is shown in fig 3 to be early but it is one of the last sections to explain.
- Sections 3.1 and 3.1 refer to figure 4 in order of b then a and c. Consider reordering your figure to make it more organized
- Figure 4 b needs to have a labelling of what the colors mean

Results:
- Table 3 doesn't show fair comparisons where there are only selected comparisons which have multiple factors changed at a time so it is hard to know what to compare with what
- Label the meaning of the gray shading in row 3 of table 3 in the caption or remove it. In table 4 there are 2 rows shaded which I don't know if it means anything or is just a stylistic choice


Additional analyses expected:
- More quantitative results with different blur levels to see when performance breaks down
- Failure modes. Any paper should explore where their model fails and try to explain that. I have tested multiple previous algorithms on novel image sets and they usually fail to generalize despite the impressive results presented in papers which makes me skeptical of these types of papers.
- To check for generalization, you should test on out-of-distribution artificial images as even natural images from different categories or different datasets are still close distribution. Check: https://arxiv.org/abs/2405.10078 where they show that despite the wealth of images in some datasets, they are insufficient to generalize

**Strengths And Weaknesses:**

Strengths:
- The model is smaller than previous ones
- It achieves comparable or better performance in the tested conditions

Weaknesses:
- The paper is hard to read with many sections appearing to be shuffled instead of following a logical sequence.
- The experiments are insufficient and there are no limitations and failure modes section.
- Figures are hard to see
-

---

> ### Author Response · Authors · 2025-03-31
> **Response to Reviewer vkx1 of Paper3985**
>
> We have carefully read the reviewer's comments and sincerely appreciate their advice on revising our paper. We acknowledge that the writing was rushed, and we are making great efforts to improve it based on Requested Changes. Thank you!

---

### Review · Reviewer_wXjn · 2025-04-02

**Summary Of Contributions:**

The paper studies the problem of image deblurring with a transformer-based neural network architecture. The paper claims contributions in network architecture design and specific module design modifications. The proposed method is shown to achieve improved deblurring quality in both quantitative and qualitative evaluation. The proposed method is also shown to have better generalization capability to various content and lighting conditions.

**Audience:**

Yes

**Broader Impact Concerns:**

I have no concern on the broader impact.

**Claims And Evidence:**

Yes

**Requested Changes:**

The authors are encouraged to make revisions according to the details in the discussion of the weaknesses. Besides, there are some minor issues:
1. p.3 2nd paragraph, change "simply" --> "simplify"?
2. p.3 "Transformer networks have been studies for the image deblurring problem"
3. p.5 "We assert the DeblurDiNAT remains a robust model"

**Strengths And Weaknesses:**

# Strength
1. The method proposed in this paper is demonstrated to significantly improve sharpness, while free from artifacts. The authors have included abundant visual examples to demonstrate the advantage of the proposed method.
2. The evaluation is conducted on various datasets, and the proposed method is shown to have good generalizability over different content and lighting conditions in different datasets.

# Weakness
My concerns of weakness in this paper is mainly regarding the presentation clarity, readability, and adequacy of details in the descriptions.
1. One of the main motivation, as discussed in the introduction section, is that existing methods sometimes produce undesirable noisy pixels regardless of the input. However, this point is somehow inconsistent with the remaining part of the paper: the authors neither explain why this happens to existing models but not the proposed, nor motivate the proposed techniques to avoid these potential glitches. Besides, the artifacts produced by other methods, as the authors show in Fig. 1, to me is more like an overflow issue of the pixel values and might be just addressed by clamping the output value. The authors should either provide more analysis on this phenomenon to actually make a point, or rephrase the introduction with a different motivation.

2. The flow of the last 4 paragraphs in the introduction is not good, hard to follow, and not showing the true merits of the proposed method. Specifically, the authors might want to address the following issues:
* These 4 paragraphs are isolated points that are not grounded to actually challenges faced by existing methods. The author may want to revise them to clearly show that each innovation is for addressing what specific problem.
* One sentence suddenly mentioned point-wise 2D convolutions, without any connections with the topic.
* Please revise " inspired by the simple gate without any non-linear activation in CNN-based NAFNet".

3. In the related work section, I think the authors should make a point why transformer-based method is better than CNN-based ones or why the authors have chosen transformers (instead of saying because everybody does it).

4. The description of the proposed method is also very unclear, with the following problems:
* Some terms are not defined, e.g. $\rho$ in Eq. (1)
* The mechanism of the dilated factor is unclear.
* Fig. 3 should immediately explain "EB", rather than explaining it in a paragraph few pages later.
* It does not make sense to me why the 1D convolution along the channel axis can work. The authors should explain.

5. Some important details are carelessly missing, e.g. what is the loss function for training?

---

### Note · Authors · 2025-04-18

**Comment:**

Thanks for the criticisms from reviewers. We appreciate your hard work!. Based on the valuable comments , we decide to withdraw the submission and refine our work. Best, the authors.

**Withdrawal Confirmation:**

I have read and agree with the venue's withdrawal policy on behalf of myself and my co-authors.